# Prevalence of diarrheal diseases and associated factors among under five children in Africa: A meta-analysis

Eshetu Abera Worede[1]*, Asmamaw Malede[1], Hailemariam Feleke[1], Geziew Abere[1], Eyayaw Addissu Demeke[2], Jember Azanaw[1]

**1** Department of Environmental and Occupational Health and Safety, Institute of Public Health, College of Medicine and Health Sciences, University of Gondar, Gondar, Ethiopia, **2** Department of Physiotherapy, Bahirdar University, Bahirdar, Ethiopia

* aberaeshetu44@gmail.com

## Abstract

### Background

Diarrheal diseases remain a major health threat for children under five in Africa, causing high rates of morbidity and mortality. The regional and continental pooled prevalence and risk factors for childhood diarrhea in Africa remain unknown. This systematic review and meta-analysis (2013–2023) aims to synthesize existing evidence to estimate the pooled prevalence and identify key risk factors associated with childhood diarrhea.

### Methods

Searches were conducted in databases including PubMed, Scopus, Web of Science, and Google Scholar to identify research published between January 2013 to December 2023. The PRISMA flowchart guideline was used to screen studies. A random-effects model was used to estimate the pooled prevalence. Publication bias was assessed using a funnel plot and Egger's test, with heterogeneity assessed by $I^2$ statistics.

### Results

From the reviewed research, 66 studies met the inclusion criteria and were included in the analysis. The pooled prevalence of diarrheal diseases was 23.59% (95% CI: 21.77–25.42). Sub-group analysis by region revealed the highest prevalence found in Southern Africa (30.14%,) followed by North Africa (28.16%), Central Africa (25.25%), East Africa (24.92%), and West Africa (15.11%). Poor handwashing practices (AOR = 4.69, 95%CI: 2.44–9.04), unimproved water sources (AOR = 1.68, 95%CI: 1.44 1.95), poor solid waste (AOR = 2.29, 95%CI:(1.64 3.20), liquid waste

**Data availability statement:** All relevant data are within the paper and its Supporting information files.

**Funding:** The author(s) received no specific funding for this work.

**Competing interests:** The authors have declared that no competing interests exist.

(AOR = 1.72, 95%CI: 1.71–1.723) management, lack of latrine (AOR = 1.17, 95% CI: 1.13–1.22), were significantly associated with DD among under-five children. Conversely, protected water sources (AOR = 0.38, 95% CI: 0.27–0.53), and ventilated improved pit latrines (AOR = 0.85, 95% CI: 0.77–0.95) were protective. Additional risk factors included age (6−11 months: AOR = 1.72, 95% CI:(1.61 1.84);12–23 months: AOR = 2.92, 95% CI:1.60 5.31), lack of exclusive breastfeeding (AOR = 3.06, 95%CI: 2.12–4.43), having more than two under-five children in a household (AOR = 1.48, 95% CI: 1.28–1.71), larger family size (AOR = 2.34, 95% CI: 1.42–3.85), Maternal or caregiver illiteracy (AOR = 2.43, 95%CI: 1.95 3.03), low socioeconomic status (AOR = 1.44, 95%CI: 1.10 1.88) were also significantly associated with DD.

## Conclusion

The pooled prevalence of diarrheal diseases among under-five children in Africa was high. Age of a child, lack of exclusive breastfeeding, having more than two children in a household, low socioeconomic status, maternal or caregiver illiteracy, poor hand washing practices, unimproved water sources, poor solid and liquid waste management, absence of a latrine were factors significantly associated with childhood diarrhea. On the other hand, having a ventilated improved pit latrine and getting water from protected sources were protective factors. These findings highlight critical areas for targeted interventions to reduce diarrheal disease in vulnerable African populations.

## Introduction

The World Health Organization reported in 2021 that diarrheal disease accounted for 8% of fatalities among children under the age of five [1]. These illnesses remain a major contributor to preventable child mortality, particularly in low-income countries [2]. Despite global health efforts, diarrheal diseases persist as a significant public health challenge in developing nations, with the burden disproportionately affecting the regions across Africa [3]. The elevated mortality rates from diarrheal diseases are largely attributed to insufficient sanitation, poor hygiene, limited access to safe drinking water, and undernutrition, which are challenges in many African countries [4,5]. These conditions facilitate the spread of diarrheal pathogens, further intensifying the health burden on at-risk populations [6,7].

In Africa, the burden of diarrheal diseases among children under five is further intensified by multiple socioeconomic and health system challenges. Poor health infrastructure, high poverty rates, and limited access to healthcare services contribute to the high incidence of diarrhea [8,9]. Repeated episodes of diarrhea can lead to severe dehydration, malnutrition, and stunted growth, which increases the vulnerability of young children to the condition [10,11]. Despite the implementation of preventive measures such as oral rehydration therapy, improved sanitation, and vaccination programs, diarrheal diseases continue to pose a major public health challenge across

the continent [11,12]. The effectiveness of these interventions is often hindered by logistical, financial, and cultural barriers, leading to disparities in health outcomes [11,13].

Previous studies have consistently reported high prevalence rates of diarrheal diseases among under-five children in Africa [14-15]. A systematic review by Walker et al. [16] highlighted that sub-Saharan Africa exhibits one of the highest incidences of childhood diarrhea globally, with significant variations between urban and rural settings [17]. Contributing factors include unsafe drinking water, inadequate sanitation facilities or latrines, poor personal hygiene, and limited breast-feeding practices [18,19]. Socio-demographic factors such as maternal education, household income, and urban-rural residence have been identified as significant determinants of diarrheal disease prevalence among children [4,8]. Research indicated that children from lower socioeconomic backgrounds are at a higher risk of experiencing frequent episodes of diarrhea [4,20]. Environmental factors, including the proximity of households to water sources and waste disposal practices, also play critical roles in the spread of diarrheal pathogens [5,21].

Various studies have explored the role of healthcare interventions in mitigating the burden of diarrheal diseases. The promotion of exclusive breastfeeding, vaccination against rotavirus, and the use of zinc supplementation have been shown to significantly reduce the incidence and severity of diarrhea in children [9,11]. However, these interventions' implementation and uptake vary widely across African countries, often due to logistical, financial, and cultural barriers [11,13]. This systematic review and meta-analysis focused on studies published between 2013 and 2023, aimed to consolidate and update the existing evidence on the prevalence of diarrheal diseases among under-five children in Africa. Despite having rich evidence on diarrheal diseases in Africa, there remains a pressing need for comprehensive and updated evidence on the prevalence and risk factors associated with these conditions among under-five children. This systematic review and meta-analysis aimed to synthesize existing data to provide a more accurate estimate of the burden of diarrheal diseases among under-five children in Africa, identify key risk factors, and inform future public health strategies and interventions [11,13].

## Methods and materials

### Protocol registration

The review protocol was registered with the PROSPERO database through a registration number (PROSPERO- CRD42024579271)

### Eligibility Criteria

Inclusion Criteria: Studies published between January 2013 and December 2023 with study designs of cross-sectional, cohort, and case-control studies and reporting prevalence of diarrheal diseases and/or associated factors among under-five children in African countries were included. Studies published in the English language were included. Further, studies that reported data on diarrheal disease prevalence with a clear definition of cases based on clinical or self-reported symptoms were included.

**Exclusion Criteria:** Studies that were not conducted in African countries and among under-five children were excluded. In addition, reviewed papers, editorials, commentaries, case reports, and studies without primary data and studies with incomplete data on prevalence or unclear diagnostic criteria for diarrheal diseases were excluded.

### Data sources and Search Strategy

To conduct this systematic review and meta-analysis, PubMed, Scopus, Web of Science, and Google Scholar were used as comprehensive electronic search databases. Research published from January 2013 and December 2023 was included to capture the most recent and relevant information. Keywords and medical subject headings (Mesh) used included: ("Diarrhea" OR "Diarrhoea" OR "Diarrheal Diseases") AND ("Prevalence" OR "Incidence" OR "Epidemiology" OR "Burden") AND ("Under-Five Children" OR "Children under five" OR "Children under 5 years" OR "Preschool children" OR

"Infants") AND ("Africa" OR "Sub-Saharan Africa" OR "East Africa" OR "West Africa" OR "Central Africa" OR "North Africa" OR "Southern Africa").We followed the PRISMA 2020 checklist for reporting (S1 Table). Reference lists of relevant articles were also manually searched to include additional eligible studies (S1 File).

## Study design and selection processes

This systematic review and meta-analysis were conducted to estimate the pooled prevalence of diarrheal diseases and determinant factors among under-five children in Africa. This review followed the Preferred Reporting Items for Systematic Reviews and Meta-Analyses (PRISMA 2020 checklist) protocol to ensure transparency and reproducibility of the findings (Fig 1). Two independent reviewers (EAW and JA) screened the titles, abstracts, and full text for all identified articles. In the screening phase, studies whose titles and abstracts were inadequate to meet the eligibility criteria were removed. Full texts of relevant studies were then retrieved and assessed for eligibility for the final meta-analysis. Any discrepancies between reviewers to include and exclude articles were resolved through discussion of the team based on the inclusion and exclusion criteria.

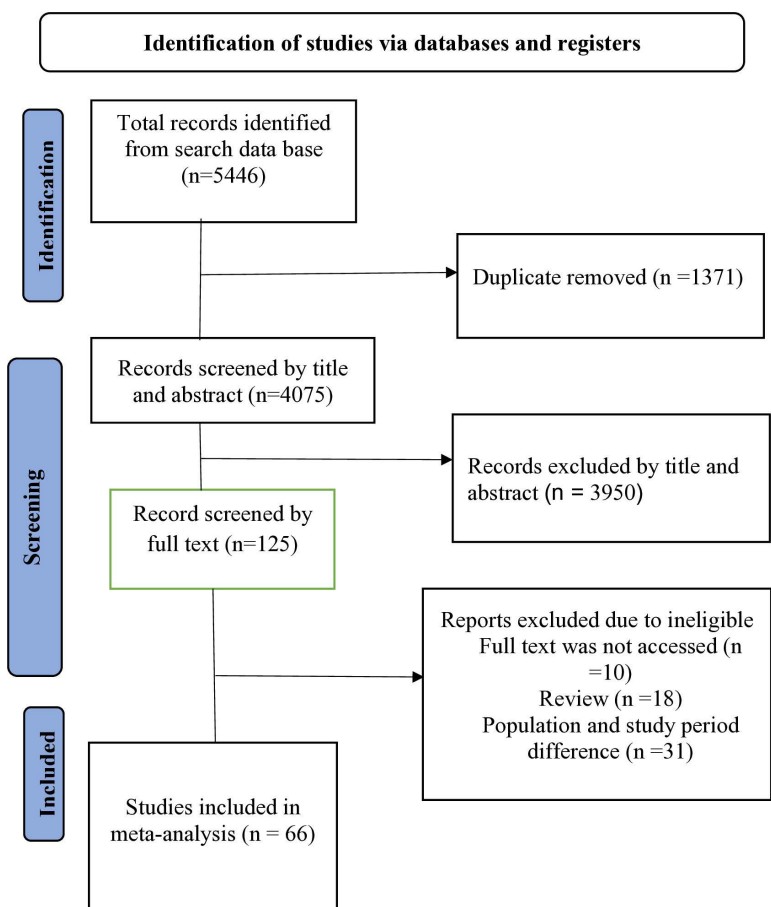

**Fig 1. PRISMA flow diagram for the systematic review and meta-analysis of DD and associated factors among under-five children in Africa (2013 −2023).**

## Data extraction and compilation

Two authors (EAW and JA) independently extracted data using a Microsoft Excel spreadsheet from June 01 to July 30, 2024. The components of extracted data included the following characteristics as Author(s) and years of publication, Country and setting, region, study design and sample size, research title, prevalence, and risk factors (odds ratio with its 95%CI) of included studies to assess pooled prevalence of diarrhea and pooled odds ratio of determinant factors (Table 1) and (S2 File).

## Quality assessment

The methodological quality of the included studies was assessed using a modified version of the Newcastle–Ottawa Scale (NOS) [82,83]. Two independent reviewers(EAW and JA) assessed each study's qualities, including the clarity of the research aims, appropriateness of the methodology and research design, recruitment strategy, data collection method, researcher-participant relationship, ethical considerations, data analysis, statement of findings, and overall value of the research. Discrepancies during the evaluation process were resolved through discussion among the reviewers. The methodological quality of each study included in the analysis was assessed using a rating system such as very good (9–10 points), good (7–8 points), satisfactory (5–6 points), or unsatisfactory (0–4 points). Studies with a score of ≥ 7 out of 10 on this scale were deemed to have achieved high methodological quality [84]. Consequently, only studies aligned within the good and very good quality categories' as per the criteria were considered for final analysis (S3 file). It is noteworthy that studies rated as very good quality, indicating a higher level of methodological rigor, were given special attention and were ultimately included in the conclusive analysis.

## Outcome variable

In this review, there were two outcomes of interest, one was the pooled prevalence of diarrheal disease among under five children, and the second was a pooled measure of the association between DD and associated factors among under five children in Africa which was estimated using the pooled odds ratio (OR) with a 95% confidence interval.

## Statistical analysis

The data extracted in a Microsoft Excel spreadsheet was exported to STATA version 17 software for further analysis. To estimate the pooled effect size with 95% CI for each study, a random-effect model meta-analysis was used. Heterogeneity and Publication bias were assessed using the $I^2$ statistic ($I^2$ =above 50% indicated substantial heterogeneity), funnel plots, and Egger's test (p > 0.05 indicated no publication bias), respectively. A meta-regression analysis was also done to quantify additional publication bias. Sensitivity analyses were conducted to explore the robustness of the findings by excluding low-quality studies and studies with extreme prevalence estimates.

# Results

## Characteristics of included studies

A total of 260,226 study participants from 66 articles were included in this systematic review and meta-analysis. Almost all studies (60 studies) were done using cross-sectional design and others were done through case-control (2), mixed methods (2), and prospective cohort (2) design. Regarding the study setting, 56 articles were done in community-based, 8 in a population survey, 1 in a health institution, and 1 among internally displaced persons (IDPs). Of the total included studies, thirty-nine (39) were in East Africa, 14 in West Africa, 3 in North Africa, 7 in South Africa, and 3 in Central Africa (Table 1). According to NOS quality assessment criteria, all included studies had met the specified quality with a quality score ≥ 7 (S3 File).

## Prevalence of diarrheal disease among under-five children in Africa

The pooled prevalence of diarrheal disease among under-five children in Africa was found 23.59% (95% CI: 21.77–25.42; I2 = 99.3%) (Fig 2). The lowest prevalence of diarrheal disease was found in Nigeria,7.5% (95% CI: 5.76–9.24) [46], and

**Table 1. Characteristics and collected information from included studies (n = 266,450).**

| Author, year of publication, and Reference | Design | Setting | Country | Region | SS | prevalence | Quality Score |
|---|---|---|---|---|---|---|---|
| Tadesse et al. (2022) [22] | CS | CB | Ethiopia | EA | 556 | 26.8 | 9 |
| Getahun, et al. (2021) [23] | CS | CB | Ethiopia | EA | 485 | 17.6 | 9 |
| Mohammed et al. (2013) [24] | CS | CB | Ethiopia | EA | 590 | 30.5 | 9 |
| Natnael et al. (2021) [25] | CS | CB | Ethiopia | EA | 340 | 11 | 8 |
| Girmay et al. (2023) [26] | CS | CB | Ethiopia | EA | 1,807 | 14.8 | 9 |
| Bitew et al. (2023) [27] | CS | CB | Ethiopia | EA | 422 | 24.9 | 8 |
| Godana et al. (2013) [28] | CC | CB | Ethiopia | EA | 612 | 24 | 9 |
| Nantege et al. (2022) [14] | CS | CB | Uganda | EA | 384 | 62.4 | 7 |
| Asgedom et al. (2023) [29] | CS | CB | Ethiopia | EA | 4,381 | 25.5 | 8 |
| Thiam et al. (2017) [30] | CS | CB | Senegal | WA | 596 | 26 | 9 |
| Omona et al. (2020) [31] | CS | CB | Uganda | EA | 244 | 29.1 | 9 |
| Mulatu et al. (2022) [32] | PC | PS | Ethiopia | EA | 6,261 | 41.75 | 9 |
| Birhan et al. (2023) [3] | CS | CB | Ethiopia | EA | 717 | 29 | 9 |
| Workie et al. (2019) [33] | CS | CB | Ethiopia | EA | 614 | 32.1 | 9 |
| Gizaw et al. (2017) [34] | CS | CB | Ethiopia | EA | 367 | 31.3 | 8 |
| Melese et al. (2023) [35] | CS | CB | Ethiopia | EA | 400 | 20.8 | 8 |
| Anteneh et al. (2017) [36] | CS | CB | Ethiopia | EA | 775 | 21.5 | 9 |
| Gebru et al. (2014) [37] | CS | CB | Ethiopia | EA | 792 | 16 | 8 |
| Mihrete et al. (2014) [38] | CS | CB | Ethiopia | EA | 925 | 22.05 | 9 |
| McClelland et al. (2022) [39] | CS | CB | Tanzania | EA | 779 | 32.1 | 7 |
| Woldu et al. (2016) [40] | CS | CB | Ethiopia | EA | 704 | 26.1 | 8 |
| Hashi et al. (2016) [41] | CS | CB | Ethiopia | EA | 498 | 14.5 | 9 |
| Alemayehu M. et al. (2020) [42] | CS | CB | Ethiopia | EA | 722 | 23.5 | 9 |
| Siziya et al. (2013) [43] | CS | CB | N.Sudan | NA | 23,295 | 28.2 | 9 |
| Wasihun et al. (2018) [44] | CS | CB | Ethiopia | EA | 610 | 27.2 | 9 |
| Machava et al. (2022) [45] | CS | CB | Mozambique | SA | 300 | 33.3 | 9 |
| Akinyemi et al. (2022) [46] | CS | CB | Nigeria | WA | 882 | 7.5 | 8 |
| Asfaha et al. (2018) [47] | CC | CB | Ethiopia | EA | 600 | 14.18 | 8 |
| Tambe et al. (2015) [48] | CS | CB | Cameroon | CA | 602 | 23.8 | 9 |
| Mengistie et al. (2013) [49] | CS | CB | Ethiopia | EA | 530 | 14.6 | 8 |
| Melese et al. (2019) [50] | CS | CB | Ethiopia | EA | 537 | 13.6 | 9 |
| Soboksa et al. (2021) [51] | CS | CB | Ethiopia | EA | 9,916 | 11 | 7 |
| Kefalew A. et al. (2021) [52] | CS | CB | Ethiopia | EA | 620 | 24 | 9 |
| Dagnew et al. (2019) [53] | CS | CB | Ethiopia | EA | 498 | 14.5 | 9 |
| Nwokoro et al. (2020) [54] | CS | CB | Nigeria | WA | 469 | 10.77 | 9 |
| Diouf et al. (2014) [55] | CS | CB | Burundi | EA | 903 | 32.6 | 8 |
| Colombo et al. (2023) [56] | CS | CB | Côte d'Ivoire | WA | 567 | 27 | 8 |
| Danquah et al. (2014) [57] | CS | CB | Ghana | W A | 378 | 13 | 9 |
| Tampah-Naah (2019) [58] | CS | CB | Ghana | WA | 4,821 | 18 | 9 |
| Abu et al (2018) [59] | CS | CB | Ghana | WA | 401 | 18 | 7 |
| Apanga et al. (2021) [60] | CS | PS | Ghana | WA | 8,879 | 17 | 8 |
| Raza et al. (2020) [61] | CS | CB | Mozambique | SA | 13,745 | 11.1 | 9 |
| Guillaume et al. (2020) [62] | CS | CB | Kenya | EA | 324 | 18.7 | 9 |
| Fagbamigbe et al. (2017) [63] | CS | CB | Nigeria | WA | 13,322 | 13 | 9 |
| Onyearugha et al. (2020) [64] | PC | HI | Nigeria | WA | 890 | 11.2 | 9 |
| Yaya et al. (2018) [5] | CS | CB | Nigeria | WA | 28,596 | 11.3 | 9 |

*(Continued)*

**Table 1.** (Continued)

| Author, year of publication, and Reference | Design | Setting | Country | Region | SS | prevalence | Quality Score |
|---|---|---|---|---|---|---|---|
| Ntshangase et al. (2022) [21] | CS | CB | South Africa | SA | 506 | 67.3 | 8 |
| Birungi et al. (2016) [65] | CS | CB | Uganda | EA | 367 | 33 | 8 |
| Kapwata et al. (2018) [66] | CS | CB | South Africa | SA | 408 | 20 | 9 |
| Claudine et al. (2021) [20] | CS | PS | Rwanda | EA | 7,474 | 12.7 | 8 |
| Bennion et al. (2021) [67] | CS | CB | Tanzania | EA | 9,996 | 23.2 | 9 |
| Rukambile et al. (2023) [68] | MM | CB | Tanzania | EA | 340 | 22.2 | 7 |
| Bah et al. (2022) [69] | CS | CB | Sierra Leone | WA | 1,002 | 12.3 | 9 |
| Atari et al. (2023) [70] | CS | PS | South Sudan | EA | 8,338 | 19 | 9 |
| Daffe et al. (2022) [71] | CS | CB | Senegal | WA | 10,851 | 18.16 | 9 |
| Moon et al. (2019) [72] | CS | CB | Malawi | SA | 14,872 | 20 | 9 |
| Awoniyi et al. (2021) [73] | CS | PS | Nigeria | WA | 30,068 | 12.9 | 8 |
| Elmanssury et al. (2022) [74] | CS | CB | N.Sudan | NA | 311 | 35 | 8 |
| Netsereab et al. (2017) [75] | CS | PS | N.Sudan | NA | 14,081 | 26.9 | 9 |
| Jayte et al. (2023) [76] | CS | IDPs | Somalia | EA | 318 | 16.7 | 8 |
| Chilambwe et al. (2015) [77] | CS | CB | Zambia | SA | 718 | 44.6 | 9 |
| Oyedele et al. (2023) [78] | CS | PS | Zambia | SA | 4,786 | 16.88 | 7 |
| Dharod et al. (2021) [79] | CS | PS | Cameroon | CA | 2,129 | 32.3 | 8 |
| Lanyero et al. (2021) [80] | CS | CB | Uganda | EA | 856 | 37.1 | 9 |
| Aderinwale et al. (2023) [81] | CS | PS | Chad | CA | 16,710 | 19.7 | 9 |
| William et al. (2023) [15] | MM | CB | South Sudan | EA | 439 | 64.2 | 9 |

Where SC; cross- sectional, CC; case-control, PC = prospective cohort, MM = mixed method, CB = community based, PS = population survey, HI = Health institute, IDPs = internally Displaced person, EA = East Africa, SA = South Africa, A = Central Africa, NA = Northern Africa, WA = West Africa.

the highest prevalence of diarrheal disease was found in South Africa, 67.3% (95% CI: 63.21–71.38) [21], South Sudan, 64.2% (95% CI:59.71–68.68) [15], Uganda, 62.4% (95% CI:57.55–67.25) [14] and Ethiopia, 41.75% (95% CI:40.53–42.97) [32].

## Subgroup analysis of the pooled prevalence of diarrheal diseases in different African Regions

Due to higher significant heterogeneity in included studies ($I^2$ = 99.3%, P = 0.0001), subgroup analysis was conducted to determine the pooled prevalence of diarrheal disease by region. The higher and lower heterogeneity was found in southern Africa ($I^2$ = 99.6, P = 0.0001) and North Africa ($I^2$ = 86.4, P = 0.001), respectively (Table 2). Subgroup analysis by region revealed the highest prevalence in Southern Africa (30.14%, 95% CI: 23.37–36.90), followed by North Africa (28.16%, 95% CI: 26.52–29.79), Central Africa (25.25%, 95% CI: 16.40–34.11), East Africa (24.92%, 95% CI: 21.82–28.03), and West Africa (15.11%, 95% CI: 13.46–16.76) (Fig 2). There was a significant difference among these regions (p-value < 0.001), indicating that each region exhibits distinct DD prevalence.

## Heterogeneity and publication bias

The heterogeneity and publication bias were assessed within the included studies. Publication bias was assessed using a funnel plot and Egger's test, which indicated visually the funnel plot is asymmetry and Egger's test was significant (p = 0.001), suggesting publication bias (Fig 3). To adjust for this bias, we applied the trim and fill method, which showed a slight reduction in the overall prevalence estimate. In addition, due to the included studies showing a high degree of heterogeneity (I2 = 99.3%), meta-regression analysis was conducted using publication year, sample size, region,

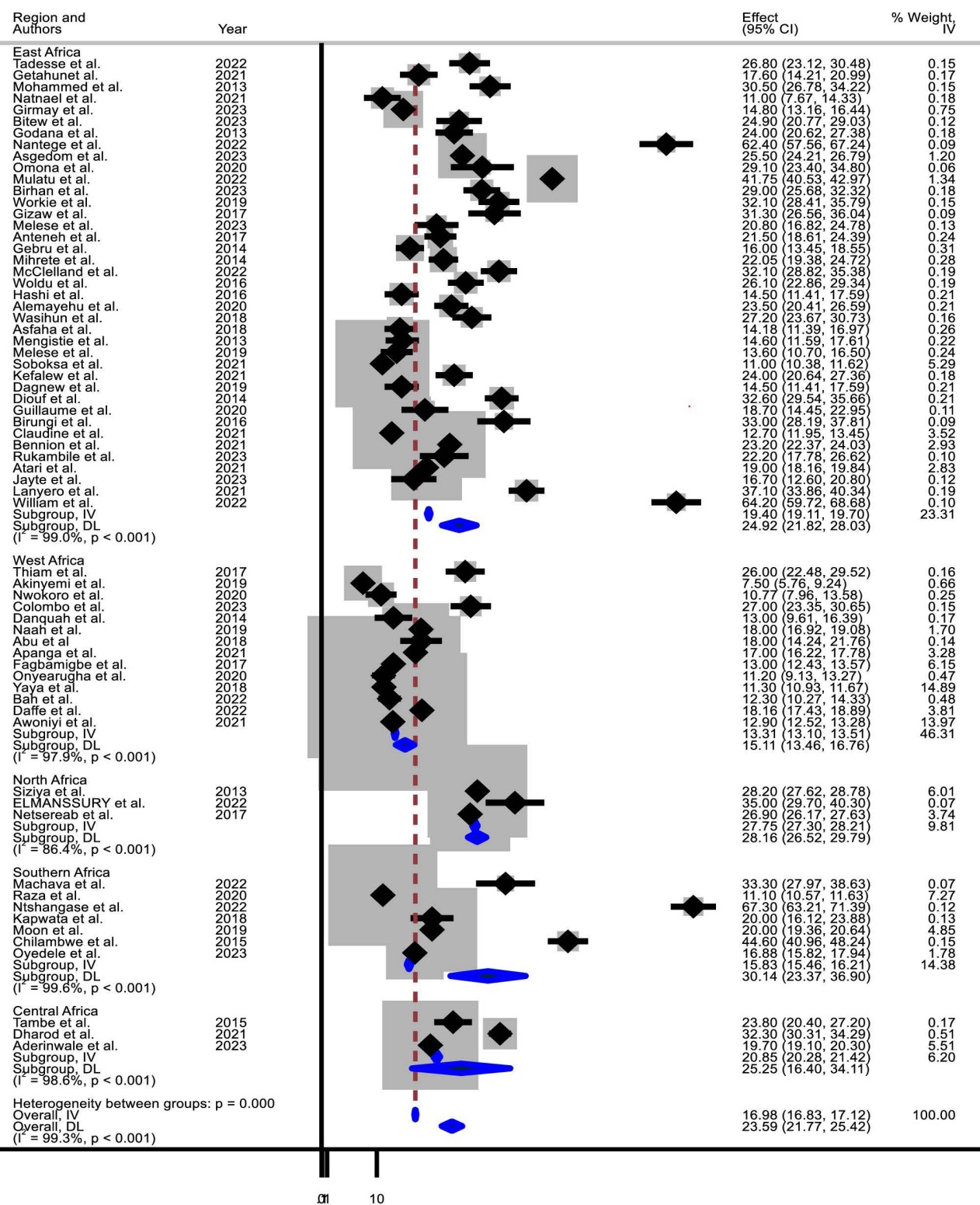

**Fig 2. Forest plot of pooled prevalence of DD among under-five children in Africa and Subgroup analysis by region (2013-2023).**

**Table 2. This table indicated that Heterogeneity among the included studies.**

**Cochran's Q statistics for heterogeneity**

| Measure | value | df | p-value | I² |
|---|---|---|---|---|
| East Africa | 3787.69 | 38 | 0.000 | 99.0% |
| West Africa | 623.00 | 15 | 0.000 | 97.6% |
| North Africa | 14.69 | 2 | 0.001 | 86.4% |
| Southern Africa | 1372.31 | 6 | 0.000 | 99.6% |
| Central Africa | 144.50 | 2 | 0.000 | 98.6% |
| Overall | 9920.81 | 67 | 0.000 | 99.3% |

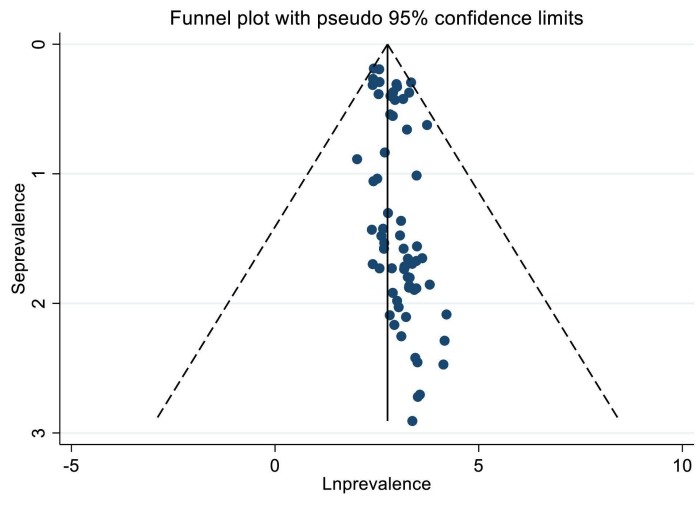

Number of studies = 66    Root MSE    = 0.5692

| Std_Eff | Coefficient | Std. err. | t | P>|t| | [95% conf. interval] |
|---|---|---|---|---|---|
| slope | 2.612518 | 0.059639 | 43.81 | 0.000 | 2.493375    2.73166 |
| bias | 0 .3505238 | 0.3505238 | 3.45 | 0.001 | 0.1475332    0.5535144 |
| Test of H0: no small-study effects | | P = 0.001 | | | |

**Fig 3. This figure indicates the funnel plot and Eggers test for publication bias assessment of the pooled prevalence of DD among under-five children in Africa.**

and standard error as covariates to identify potential sources of variation. The analysis found no significant association between DD prevalence and these covariates (p = 0.200 for publication year, p = 0.45 for sample size, p = 0.066 for region). This suggests that the observed heterogeneity was not influenced by region, sample size, and year of publication (Table 3).

## Sensitivity analysis test

Sensitivity analyses were conducted to evaluate the robustness of the meta-analysis results. Leave-one-out analysis showed that excluding individual studies did not significantly alter the overall effect size. The effect size and confidence intervals for most omitted estimates overlap substantially with the combined confidence interval (13.18, 19.57), suggesting

**Table 3.  Meta-regression to assess between-study variation (heterogeneity) among included studies.**

| Source of Heterogeneity | Coefficient | Std. err. | t | P > |t| | [95% CI |
|---|---|---|---|---|---|
| Year of Publication | −0.0373738 | 0.0288351 | −1.30 | 0.200 | −0.0950144 0.0202668 |
| Sample Size | −6.57e-06 | 8.69e-06 | −0.76 | 0.452 | −0.0000239 0.0000108 |
| Region | −0.1127427 | 0.0601661 | −1.87 | 0.066 | −0.233013 0.0075275 |
| Constant | 78.76346 | 58.27115 | 1.35 | 0.181 | −37.7189 195.2458 |

that no single study significantly alters the overall estimate. Therefore, the overall pattern indicates that this meta-analysis is stable, and the combined effect size is not overly sensitive to the inclusion or exclusion of any single study (Fig 4).

### Risk factors associated with Diarrheal disease among under-five children in Africa

In this meta-analysis, factors associated with diarrheal disease (DD) were identified based on the pooled effects of different studies. Fourteen significant pooled factors were identified associated with diarrheal disease, with each factor supported by a minimum of two articles, and a maximum of twenty studies (S1 Fig-S14 Fig) The pooled effects of water, sanitation, and hygiene (WASH) related factors were significantly associated with DD among under-five children in Africa.

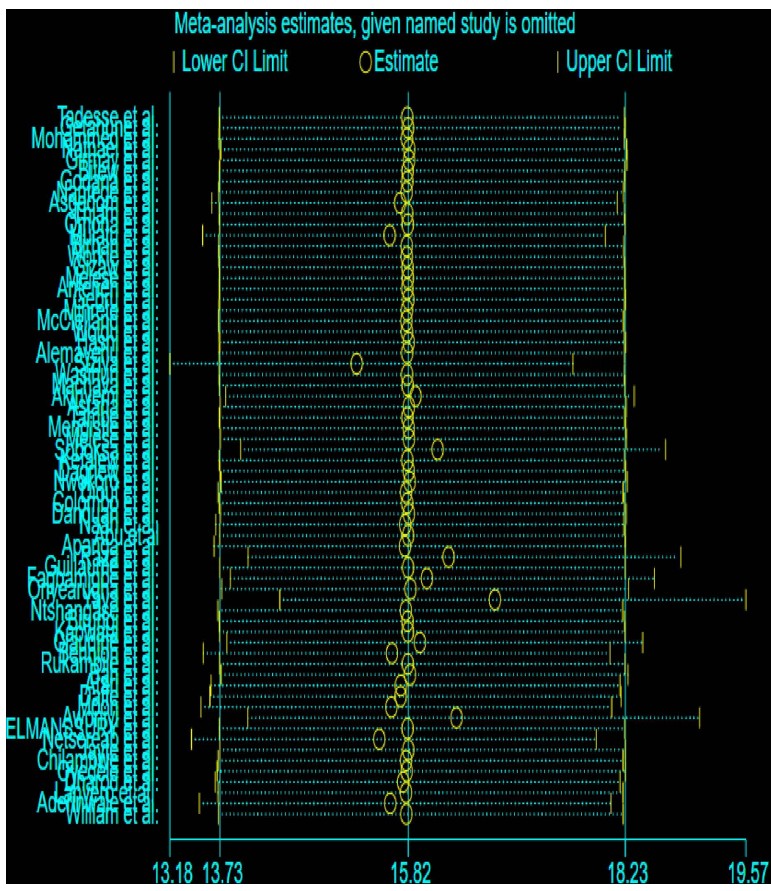

**Fig 4.  Sensitivity analysis to examine each study's effect on pooled DD prevalence among under-five children in Africa.**

The pooled effects of nine studies [14,24,27,34,37,44,50,53,63] confirmed that poor handwashing practices were significantly associated with DD in children (pAOR = 4.69, 95% CI: 2.44–9.04). In addition, the pooled effect of twenty [3,5,26,28,29,32,33,35,41,42,44,46,47,58,63,65,66,69,71,79] revealed that children who drank from unimproved water sources had a 1.68 times higher likelihood of having DD (pAOR = 1.68, 95% CI:1.44 1.95) compared to children's get water source from improved sources. On the other hand, under-five children who used protected water sources [14,31,81] had a 71% lower risk of DD when compared to those who used unprotected water sources (pAOR = 0.29, 95% CI: 0.18–0.48). Furthermore, the pooled results from eight [3,29,30,37,44,47,49,50] and nine studies [5,26,29,30,33,41,42,52,63] confirmed that poor solid waste (pAOR = 2.29, 95% CI: (1.64 3.20) and liquid waste (pAOR = 1.72, 95% CI: 1.71–1.723) management were significantly associated with DD respectively. Moreover, the combined effects of ten studies [28,33,35,38,41,42,47,63,72,79] revealed that lack of latrine was a factor for DD (pAOR = 1.17, 95% CI: 1.13–1.22). On the other hand, under-five children from families with a VIP latrine [14,32] had a 15% lower risk of DD (pAOR = 0.85, 95% CI: 0.77–0.95).

In addition, WASH-related factors, socio-demographic factors, and child-feeding practice-related factors were identified as being associated with DD. The pooled effects of nine studies [34,40,43,47,49,52,58,60,75] found that children aged between 6–11 months were 1.72 times more likely to have diarrheal disease compared to lower age categories (pAOR = 1.72, 95% CI: (1.61 1.84). In addition, children aged between 12–23 months [25,43,47,51,60,67,75] were also more likely to develop diarrheal disease compared to children below sex months or higher age categories (pAOR = 2.92, 95% CI: (1.60 5.31). Furthermore, a meta-analysis of three studies [26,47,53] found that children who were not exclusively breastfed had 3.06 times higher odds of developing DD compared to those who received exclusive breastfeeding (pAOR = 3.06, 95% CI: 2.12–4.43).

Moreover, the pooled findings of five studies [38,40,49,51,52] showed that the likelihood of DD was 1.48 times higher among participants having more than two under-five children compared to participants with fewer children (pAOR = 1.48, 95% CI: 1.28–1.71). Additionally, the pooled results of ten studies [24,26,34,35,37,38,40,41,47,50] indicated that children whose caregivers or mothers had no formal education were 2.43 times more likely to develop DD when compared to those children whose caregivers had formal education (pAOR = 2.43, 95% CI: (1.95 3.03). Further, children from low socioeconomic families [26,27,37,60,63,67,73,75,78] were 1.44 times more likely to have DD compared to children from families with better socioeconomic status (pAOR = 1.44, 95%CI:(1.10 1.88). Moreover, a meta-analysis of six studies indicated that children with more families [14,26,31,47,51,69] had significantly higher odds of developing DD compared to those with small family members (pAOR = 2.34, 95% CI: 1.42–3.85) (Table 4) and (S1 Fig-S14 Fig).

## Discussion

Diarrheal diseases remain a major public health challenge among under-five children in Africa. This study updates evidence on the prevalence and risk factors of diarrheal diseases over the past decade, providing continental and regional pooled estimates. Identifying key determinants is crucial for developing effective prevention strategies. The findings of this systematic review and meta-analysis reveal a high burden of diarrheal disease among children under five in Africa, with a pooled prevalence of 23.59%. This estimate indicates that nearly one in four children in this age group is affected by diarrheal disease, highlighting a significant public health concern across the continent.

Sub-group analysis by region revealed significant variations in the prevalence of DD with Southern Africa having the highest prevalence (30.14%), followed by North Africa (28.16%), Central Africa (25.25%), East Africa (24.92%) and West Africa (15.11%), which had the lowest prevalence. These disparities may be attributed to regional differences in water, sanitation, and hygiene (WASH) infrastructure, healthcare access, and socioeconomic conditions of the region.

The pooled prevalence of DD in this analysis was in agreement with a systematic review and meta-analysis conducted in Ethiopia, which reported a 22% prevalence of DD among children under five [85]. This finding is also in line with a meta-analysis conducted in India 21.70% [85], and a demographic health survey report of five Southeast Asian countries

**Table 4. Pooled effects of associated factors with diarrheal disease among children in Africa (2013-2023).**

| Factors | Authors, I2 with a p-value | AOR (95% CI) | % Weight |
|---|---|---|---|
| Poor hand-washing practice | Mohammed et al | 2.33 (1.53, 3.64) | 12.87 |
| | Bitew et al. | 8.37 (3.12 22.48) | 10.38 |
| | Nantege et al. | 4.64 (1.91 11.30) | 8.77 |
| | Gizaw et al. | 24.94 (6.68 93.11) | 8.77 |
| | Gebru et al. | 7.40 (2.61 20.97) | 10.11 |
| | Wasihun et al. | 3.71 (2.41 5.69) | 12.84 |
| | Melese et al. | 6.41 (2.51 16.38) | 10.63 |
| | Dagnew et al. | 6.10 (2.10 17.73) | 9.99 |
| | Fagbamigbe et al. | 1.15 (1.04 1.27) | 13.54 |
| | Overall, IV(I² =92.5%,P=0.001) | 4.69(2.44 9.04) | 100.00 |
| Poor solid waste Management(Yes) | Asgedom et al. | 1.30 (1.05 1.61) | 19.00 |
| | Thiam et al. | 2.07 (1.20 3.56) | 13.37 |
| | Birhan et al. | 2.13 (1.25 3.62) | 13.55 |
| | Gebru et al. | 3.19 (1.89 5.38) | 13.70 |
| | Wasihun et al. | 12.81 (2.50 65.63) | 3.49 |
| | Asfaha et al. | 2.29 (1.53 3.43) | 15.82 |
| | Mengistie et al. | 2.22 (1.21 4.07) | 12.30 |
| | Melese et al. | 2.23(1.37 7.61) | 8.77 |
| | Overall, IV (I2=69.9%,P=0.002) | 2.29(1.64 3.20) | 100.00 |
| Poor liquid waste management(Yes) | Girmay et al. | 2.74 (1.60 4.68) | 0.00 |
| | Asgedom et al. | 1.72 (1.71 1.723) | 99.94 |
| | Workie et al. | 3.86 (2.38 6.26) | 0.00 |
| | Hashi et al. | 2.03 (1.35 3.05) | 0.00 |
| | Alemayehu M. et al. | 2.31 (1.26 4.24) | 0.00 |
| | Kefalew A. et al. | 3.73 (1.91 7.29) | 0.00 |
| | Fagbamigbe et al. | 1.20 (1.05 1.37) | 0.02 |
| | Yaya et al. | 1.14 (1.04 1.25) | 0.03 |
| | Thiam et al. | 2.07(1.20 3.56) | 0.00 |
| | Overall, IV (I2=93.6%, P=0.000 | 1.720 (1.71 1.723) | 100.00 |
| Drinking water from an unimproved source | Girmay et al | 8.16 (1.69 39.43) | 0.85 |
| | Godana et al | 1.98 (1.43 2.75) | 6.55 |
| | Asgedom et al | 1.42 (1.22 1.66) | 8.54 |
| | Mulatu et al | 1.66 (1.27 2.17) | 7.24 |
| | Birhan et al | 2.36 (1.26 4.42) | 3.63 |
| | Workie et al | 2.68 (1.54 4.67) | 4.18 |
| | Melese et al | 4.60 (1.19 17.78) | 1.12 |
| | Hashi et al | 1.60 (1.14 2.24) | 6.42 |
| | Alemayehu M. et al | 1.81 (1.14 2.88) | 5.03 |
| | Wasihun et al | 3.69 (2.03 6.70) | 3.85 |
| | Akinyemi et al | 2.57 (1.23 5.40) | 2.92 |
| | Asfaha et al. | 1.83 (1.12 2.99) | 4.77 |
| | Tampah-Naah | 0.73 (0.58 0.93) | 7.68 |
| | Fagbamigbe et al. | 1.10 (1.02 1.19) | 9.16 |
| | Yaya et al. | 1.16 (1.06 1.26) | 9.10 |
| | Birungi et al. | 3.23 (1.32 7.9) | 2.20 |
| | Kapwata et al. | 2.75 (1.13 6.71) | 2.23 |

*(Continued)*

| Factors | Authors, I2 with a p-value | AOR (95% CI) | % Weight |
|---|---|---|---|
| | Bah et al. | 1.91 (1.01 3.62) | 3.54 |
| | Daffe et al. | 1.59 (1.48 1.71) | 9.19 |
| | Dharod et al. | 2.97(1.06 8.35) | 1.77 |
| | Overall, IV(I² =86.3%, p=0.001) | 1.68(1.44 1.95) | 100.00 |
| Lack of latrine | Godana et al. | 2.43 (1.20 4.92) | 0.31 |
| | Workie et al. | 4.80 (2.39 9.62) | 0.32 |
| | Melese et al. | 3.90 (1.39 10.87) | 0.15 |
| | Mihrete et al. | 3.50 (2.38 5.15) | 1.03 |
| | Hashi et al. | 4.16 (2.94 5.88) | 1.27 |
| | Alemayehu M. et al. | 2.77 (1.66 4.63) | 0.58 |
| | Asfaha et al. | 2.10 (1.34 3.29) | 0.76 |
| | Fagbamigbe et al. | 1.09 (1.04 1.14) | 72.9 |
| | Moon et al. | 1.19 (1.09 1.29) | 20.99 |
| | Dharod et al. | 1.51(1.12 2.04) | 1.69 |
| | Overall, IV(I2=93.4%,P=0.000) | 1.17(1.13 1.22) | 100.00 |
| Protected water source(yes) | Nantege et al. | 0.26 (0.11 0.65) | 31.76 |
| | Omona et al. | 0.32 (0.16 0.66) | 48.67 |
| | Aderinwale et al. | 0.26 (0.08 0.82) | 19.56 |
| | Overall, IV (I2=0.0%,P=0.926) | 0.29(0.18 0.48) | 100.00 |
| Ventilated Improved Pit latrine(Yes) | Nantege et al. | 0.50 (0.28 0.90) | 3.39 |
| | Mulatu et al. | 0.87(0.78 0.97) | 96.61 |
| | Overall, IV (I2=69.6%,P=0.004) | 0.85(0.77 0.95) | 100.00 |
| Exclusive breastfeeding(No) | Girmay et al. | 2.83(1.74 4.60) | 58.20 |
| | Asfaha et al. | 4.84(2.21 10.60) | 22.28 |
| | Dagnew et al. | 2.30(1.00 5.31) | 19.52 |
| | Overall, IV(I²=0.0%,p=0.000) | 3.06(2.12 4.43) | 100.00 |
| Children's Age(6–11 month) | Gizaw et al. | 6.28 (3.00 13.13) | 0.79 |
| | Woldu et al. | 4.80 (2.2 10.88) | 0.64 |
| | Siziya et al. | 1.47 (1.36 1.58) | 70.82 |
| | Asfaha et al. | 7.48 (2.40 23.32) | 0.33 |
| | Mengistie et al. | 2.25 (1.50 3.37) | 2.66 |
| | Kefalew A. et al. | 1.546(1.07 2.24) | 3.16 |
| | Tampah-Naah | 3.34 (2.42 4.61) | 4.13 |
| | Apanga et al. | 2.06 (1.45 2.92) | 3.53 |
| | Netsereab et al. | 2.48(2.08 2.97) | 13.93 |
| | Overall, IV(I2=89.5%,P=0.000) | 1.72(1.61 1.84) | 100.00 |
| Children's Age(12–23 month) | Natnael et al | 4.68 (4.60 4.76) | 15.87 |
| | Siziya et al | 1.52 (1.42 1.63) | 15.84 |
| | Asfaha et al | 11.59 (3.06 43.87) | 8.90 |
| | Soboksa et al | 1.49(1.11 1.98) | 15.31 |
| | Apanga et al | 2.37 (1.67 3.36) | 115.06 |
| | Claudine et al | 4.51 (3.05 6.69) | 14.85 |
| | Netsereab et al | 2.45 (1.45 4.13) | 14.17 |
| | Overall, IV(I²=99.4%,P=0.001) | 2.92(1.60 5.31) | 100.00 |

*(Continued)*

**Table 4.** (Continued)

| Factors | Authors, I2 with a p-value | AOR (95% CI) | % Weight |
|---|---|---|---|
| Formal education of mothers or Caregivers (No) | Mohammed et al | 1.89(1.38 2.59) | 22.76 |
| | Girmay et al | 3.15 (1.54 6.46) | 7.64 |
| | Gizaw et al | 6.61 (2.27 19.23) | 3.83 |
| | Melese et al | 3.70 (1.28 10.70) | 3.86 |
| | Gebru et al | 1.74 (1.04 2.92) | 12.56 |
| | Mihrete et al) | 1.81 (1.15 2.84) | 15.20 |
| | Woldu et al | 2.50 (1.20 5.20) | 7.38 |
| | Hashi et al | 3.02 (1.56 5.84) | 8.77 |
| | Asfaha et al | 2.88 (1.70 4.88) | 12.29 |
| | Melese et al | 3.97(1.69 9.32) | 5.71 |
| | Overall, IV($I^2$=24.6%,P=0.001) | 2.43(1.95 3.03) | 100.00 |
| Children in low Economic status | Girmay et al. | 5.91 (3.01 11.60) | 7.47 |
| | Bitew et al. | 3.68 (1.81 7.50) | 7.09 |
| | Gebru et al. | 1.75 (1.06 2.88) | 9.50 |
| | Apanga et al. | 0.58 (0.39 0.86) | 110.85 |
| | Fagbamigbe et al. | 1.37 (1.11 1.70) | 13.04 |
| | Claudine et al. | 1.59 (1.15 2.20) | 11.79 |
| | Awoniyi et al. | 1.49 (1.31 1.70) | 13.76 |
| | Netsereab et al. | 1.22(1.04 1.42) | 13.55 |
| | Oyedele et al. | 0.78(0.62 0.97) | 12.96 |
| | Overall, IV(I2=89.0%,p=0.001) | 1.44(1.10 1.88) | 100.00 |
| Having more than two under-five children | Mihrete et al. | 1.73 (1.03 2.92) | 7.24 |
| | Woldu et al. | 22.40 (2.83 177.25) | 0.46 |
| | Mengistie et al. | 1.74 (1.33 2.28) | 27.25 |
| | Soboksa et al. | 1.21(1.01 1.46) | 58.30 |
| | Kefalew A. et al. | 3.11(1.81 5.35) | 6.74 |
| | Overall, IV (I2=80.2%,P=0.000) | 1.48(1.28 1.71) | 100.00 |
| Having more family members | Girmay et al | 2.38(1.68 3.38) | 21.51 |
| | Nanteg et al | 2.22(1.18 4.18) | 17.35 |
| | Omana et al | 7.18(1.35 38.15) | 6.48 |
| | Asfaha et al | 4.05(1.91 8.59) | 15.53 |
| | Soboksa et al | 1.18(1.03 1.36) | 23.65 |
| | Bah et al | 2.50(1.17 5.32) | 15.48 |
| | Overall, IV (I2=83.5%,P=0.001) | 2.34(1.42 3.85) | 100.00 |

that DD ranged from 8.39% in the Philippines to 18.21% in Indonesia [86,87]. This similarity might be due to similar socio-economic and environmental contexts across the country. On the other hand, the prevalence of DD in this study exceeds the pooled estimates from the Demographic and Health Survey in East Africa (14.28%) [88] and 34 sub-Saharan African countries (15.3%) [4].

The present study also aimed to identify the determinants of diarrhea among children under five in Africa. The pooled effects of nine studies revealed that children aged between 6–11 months and 12–23 months faced a higher risk of developing diarrheal disease compared to lower age groups. This heightened vulnerability may stem from increased mobility as infants begin to crawl or walk, leading to greater exposure to environmental pathogens [89]. Furthermore, the introduction

of complementary feeding around six months of age may elevate the risk of infections through contaminated food and water further contributing to the disease burden [49,89].

In addition, children who were not exclusively breastfed were found to have DD compared to those who had exclusive breastfeeding practices. In support of this finding, a study from the UK indicated that exclusive and partial breastfeeding could prevent 53% and 31% of diarrhea hospitalizations respectively [90]. This finding is also in agreement with Pakistan [91] and studies in less developing countries [92]. This is because breast milk contains several disease-preventive nutrients and may help strengthen the immune system [26].

Furthermore, a meta-analysis of five and six studies revealed that households with more than two under-five children [38,40,49,51,52] and those with larger family sizes [14,26,31,47,51,69] had a higher likelihood of developing DD compared to households with fewer children and smaller family sizes. This finding is in line with a study from Pakistan [91]. This is justified by the fact that as the number of children in a household increases, parental care and attention tend to decline, making children more vulnerable to contamination [93]. Further, those with diarrheal disease can easily spread it to others in the household with more family members [91,93,94].

In this meta-analysis, the pooled results of ten studies indicated that children whose caregivers or mothers had no formal education were twofold to have DD when compared to those children whose mothers or caregivers had formal education. This might be due to education raising awareness of diarrhea transmission and prevention, improving household health and sanitation practices, and promoting positive behavior changes [34,37]. Higher maternal education is linked to better caregiving practices, including timely weaning, good hygiene habits, and appropriate child feeding, all of which support child health and development [26].

Further, this analysis showed that children from low socioeconomic families were more prone to diarrhea, which is in agreement with a systematic review and meta-analysis in India, which links diarrhea significantly with low socioeconomic status [95] and other studies in Ethiopia [37,40]. This could be because wealthier families have better access to soap, water purification methods, and toilets, reducing microbial contamination, while lower-income families lack these resources, increasing their risk of disease [40].

Furthermore, water, sanitation, and hygiene (WASH) related factors were significantly associated with DD. In this analysis, the pooled effects of ten studies showed that poor hand-washing practices were associated with DD in under five children. This finding is supported by an Asian study that children in households with hand-washing facilities using only water were significantly more likely to experience diarrhea than those with facilities equipped with both water and soap [96]. Additionally, another systematic review confirmed that hand washing with soap, improved water quality, and proper excreta disposal can reduce the risk of diarrhea by 48%, 17%, and 36%, respectively [97]. Further, a study in Pakistan reported a 59% prevalence of diarrheal disease among children under five was linked to poor hygiene practices such as caregivers seldom or never washing hands before feeding(83%), and mothers using only plain water after using the toilet (52.3%) [98]. This might be due to effective hand washing with soap being able to reduce bacterial load and reduce infection.

In addition, the pooled effects of twenty studies confirmed that children who drank from unimproved water sources had a higher likelihood of having DD compared to children's drank from proven sources. On the other hand, children who used protected water sources had a 62% lower risk of DD compared to those who used unprotected water sources. This may be due to the higher risk of fecal contamination typically found in unprotected water sources compared to protected ones, increasing the likelihood of waterborne illness [32].

Furthermore, the pooled meta-analysis of eight and nine studies confirmed that poor solid and liquid waste management was significantly associated with DD. This might be due to improper disposal of household waste can harbor infectious agents and create breeding sites for flies that spread pathogens, contaminating water, food, and utensils. This exposes children to high-risk environments and contributes significantly to diarrheal diseases among those under five. Moreover, the absence of latrine facilities was also identified as a contributing factor, as evidenced by the combined effects of multiple

studies. This finding is supported by a meta-analysis in Ethiopia that children living in households without latrine facilities were more likely to develop diarrhea than children living in households with such facilities [85]. On the other hand, having a ventilated improved latrine reduced the risk of DD, as confirmed by the pooled effects of two studies [99]. Household access to a latrine indicates sanitation conditions and helps prevent pathogen transmission through fecal contamination [100].

### Strengths and limitations of the study

This meta-analysis is comprehensive, employing various databases and following PRISMA guidelines, using a random-effects model and publication bias assessments that strengthen the reliability of the result. Quality assessment using the Newcastle–Ottawa Scale (NOS) ensures high-quality studies, while PROSPERO registration adds transparency. However, high heterogeneity between studies may limit generalizability, and reliance on observational data limits causal inference on childhood diarrheal disease risk factors. The other limitation is variations in study methodologies, and the reliance on available literature, which may not capture the full spectrum of determinants of diarrheal diseases among under-five children in Africa [84]. Lastly, the exclusion of studies published in languages other than English may result in the omission of relevant data, potentially limiting the full representation of determinants of diarrheal diseases among under-five children in Africa.

## Conclusions

The pooled prevalence of diarrheal disease among children under five in Africa was high which underscores a substantial health burden in this age group. Age of children, lack of exclusive breastfeeding, having more than two young children and more family members, low socioeconomic status, no mothers or caregivers had formal education, poor hand washing practices, unimproved water sources, poor waste management, and lack of latrines were significantly associated with childhood diarrhea. On the other hand, protected water sources and the presence of ventilated improved latrines were protective factors for childhood diarrheal. These findings highlight critical areas for targeted interventions to reduce diarrheal disease in vulnerable populations across Africa.

## Supporting information

**S1 Table. PRISMA 2020 checklist.**
(DOCX)

**S1 File. All studies identified in the literature search with inclusion and exclusion criteria.**
(XLSX)

**S2 File. All data was extracted from the primary research article for the systematic review and meta-analysis.**
(XLSX)

**S3 File. Newcastle Ottawa Scale Quality Assessment checklist for quality assessment of included studies.**
(PDF)

**S1 Fig. Forest plot of poor hand washing practice associated with diarrheal disease among under-five children.**
(TIF)

**S2 Fig. Forest plot of poor solid waste management associated with diarrheal disease among under five children in Africa.**
(TIF)

**S3 Fig. Forest plot of poor liquid waste management associated with diarrheal disease among under five children in Africa.**
(TIF)

**S4 Fig. Forest plot of drinking water from unimproved source associated with diarrheal disease among under five children.**
(TIF)

**S5 Fig. Forest plot of drinking water from protected source is a protective factor for diarrheal disease among under five children in Africa.**
(TIF)

**S6 Fig. Forest plot of lack of latrine associated with diarrheal disease among under five children in Africa.**
(TIF)

**S7 Fig. Forest plot of VIP latrine associated with diarrheal disease among under five children in Africa.**
(TIF)

**S8 Fig. Forest plot of no exclusive breastfeeding associated with diarrheal disease among under five children in Africa.**
(TIF)

**S9 Fig. Forest plot of children aged 6–11 Months associated with DD among under-five children in Africa.**
(TIF)

**S10 Fig. Forest plot of children aged 12–23 months associated with diarrheal disease among under-five children in Africa.**
(TIF)

**S11 Fig. Forest plot of no formal education of caregivers or mothers associated with diarrheal disease among under-five children.**
(TIF)

**S12 Fig. Forest plot of low socio economic status associated with diarrheal disease among under five children in Africa.**
(TIF)

**S13 Fig. Forest plot of having more than two under five children associated with diarrheal disease among under five children in Africa.**
(TIF)

**S14 Fig. Forest plot of more family members is associated with diarrheal disease among under-five children in Africa.**
(TIF)

## Acknowledgments

The authors extend their gratitude to the University of Gondar, Ethiopia, for providing office space and complimentary internet access. We also acknowledge the valuable contributions of the studies included in this systematic review and meta-analysis, which served as a foundation for our work.

## Author contributions

**Conceptualization:** Eshetu Abera Worede, Asmamaw Malede.

**Data curation:** Eshetu Abera Worede, Asmamaw Malede, Eyayaw Addissu Demeke, Jember Azanaw.

**Formal analysis:** Eshetu Abera Worede, Geziew Abere, Eyayaw Addissu Demeke, Jember Azanaw.

**Investigation:** Eshetu Abera Worede, Hailemariam Feleke, Eyayaw Addissu Demeke, Jember Azanaw.

**Methodology:** Asmamaw Malede, Hailemariam Feleke, Geziew Abere.

**Resources:** Hailemariam Feleke.

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
