## [Decision Letter · Decision Letter 0]

Dear Dr. Worede,

Thank you for submitting your manuscript to PLOS ONE. After careful consideration, we feel that it has merit but does not fully meet PLOS ONE’s publication criteria as it currently stands. Therefore, we invite you to submit a revised version of the manuscript that addresses the points raised during the review process.

We look forward to receiving your revised manuscript.

Kind regards,

Hope Onohuean, PhD

Academic Editor

PLOS ONE

4. As required by our policy on Data Availability, please ensure your manuscript or supplementary information includes the following:

Reviewers' comments:

Reviewer's Responses to Questions

**Comments to the Author**

1. Is the manuscript technically sound, and do the data support the conclusions?

Reviewer #1: Yes

Reviewer #2: No

2. Has the statistical analysis been performed appropriately and rigorously?

Reviewer #1: Yes

Reviewer #2: No

3. Have the authors made all data underlying the findings in their manuscript fully available?

Reviewer #1: Yes

Reviewer #2: Yes

4. Is the manuscript presented in an intelligible fashion and written in standard English?

Reviewer #1: Yes

Reviewer #2: No

Reviewer #1: Improve the clarity, coherence, and presentation of the manuscript by doing these recommendations:

1) Authors need to paraphrase sentences in two paragraphs since they have similarities with other articles. Similarity rates: 20%.

2) Authors need to check and recheck the data citation name and congruence.

3) Authors need to check and recheck the congruence of citations and references according to the PlosOne template.

4) Authors need English language editing service as well as punctuation service.

5) Authors need to make some paragraphs in discussion more concise.

6) Authors need to highlight the contributions of the paper in DD prevalence research area besides for intervention recommendation.

Reviewer #2: The paper is a systematic review that focuses on diarrheal disease in children under the age of five in Africa from January 2013 to December 2023 using PRISMA guidelines. However, a lot must be done to make this ready for publication.

• The result section of the abstract needs to be rewritten for clarity. Improve sentence structure for readability.

• "Using a systematic search of PubMed, Scopus, Web of Science, and Google Scholar for studies from 2013–2023, following PRISMA guidelines" Is ambiguous. Kindly rewrite this part for clarity.

• There is no consistency in the font size usage and alignment of the paper. Some of the words are merged, such as Sudan,64.2%(95%CI:59.71-68.68) (16), Uganda,62.4%(95%CI:57.55-67.25) (14) and Ethiopia ,41.75%(95%CI:40.53-42.97)(34)(Figure 2) in Prevalence of Diarrheal disease among under-five children in Africa.

• The degree of freedom in Cochran's Q statistics for heterogeneity is missing for Central Africa.

• The dataset is dominated by studies from East Africa, with comparatively fewer studies from Central, Northern, and Southern Africa. This regional imbalance may skew the findings. Discussing how this affects generalizability would strengthen the conclusions.

• The inclusion criterion limiting studies to those published in English may have excluded relevant research from French-speaking or Arabic-speaking African countries. This should be acknowledged.

• The heterogeneity (I² = 99.3%) is relatively high, which raises concern about the comparability of studies included in the review. Consider explaining how the high heterogeneity might impact the pooled prevalence estimates and associated risk factors.

• The plots are too compressed, especially the Sensitivity analysis and Funnel plots. Kindly revise all the plots to make them publication-worthy. The funnel plots suggested publication bias, which needs to be addressed.

• The Egger test was done with the number of studies to be 70, as indicated in the table, but your method alluded to only 69 studies.

**Do you want your identity to be public for this peer review?** For information about this choice, including consent withdrawal, please see our Privacy Policy

Reviewer #1: No

Reviewer #2: **Yes: ** Pelumi Oladipo

---

## [Author Response · Author response to Decision Letter 1]

7 Mar 2025

Citation & Figure 2

Incomplete reference (example: No year).

Missing reference

Missing reference

Table 1

Table 1 Example:

Atari et al. is 2021 or 2023?

William’s citation is 2022 or 2023?

Example:

49. Akinyemi YC. Spatial pattern and determinants of diarrhoea morbidity among under-five aged children in Lagos State NCHJ-.

Kefalew et al. 2021

There’s missing reference for Nwokoro 2018.

In the references, there are two (double) Nwokoro, 2020.

Missing year for Danquah 2014 reference

Missing references for Naah et al 2019

Double citations:

Thiam 2017

CAPSLOCK:

ELMANSSURY et al. (2022)(80) Please revise it to be the same as the reference: Getahun, not Getahunet.

Please revise. It should be 2023.

Please revise. It should be 2023.

Please recheck all the references, according to The PlosOne template. For this reference, the year should be: 2019.

Please add reference for Kefalew 2021

Please add the reference for Nwokoro (2018).

Please add the year

Please add the reference

Please delete one of them

Please check any capslock or other size format for all text besides this citation.

Comment accepted and revision is done, yes some references is directly taken from stata command is generated like getahunnet. But we authors correct it accordingly the comment is given.

Its corrected as 2023.

Comment Accepted and revised as William et al 2023 , both in Table 1 and figure 2.

Comment accepted and all reference is cheked and corrected

Kefalew et al I have checked and it is correct.

Dear editors, thank you for detail comments in the case of Nwokoro 2020, one reserch paper was enter twice, so we authors corrected it by removing one of them and 2018 is not correct ,it was 2020. All are was about Nwokoro 2020. So one is used for data analysis( Nwokkoro 2018 is removed).

Comment accepted and the year is added( Danquah 2014)

The comment has been accepted and the reference is added.

The comment is accepted, and one is deleted

The comment is accepted and the reference is added( Naah et al 2019)

Comment is accepted and one is deleted(Thiam, 2017)

The comment is accepted, and caps lock is corrected (Elmanssury et al. 2022)

33

9

Appendix Sub group of prevalence

Overall prevalence

Figure 2 on page 33 do not captured this data

Number of studies in Sub Group analysis Incongruence overall prevalence between figure 2 and 3

(95% CI: 21.3 - 24.8; I2 = 99.3%). Based on this estimate, the lowest prevalence of diarrheal disease was found in Nigeria, 7.5% (95% CI: 5.76-9.24) (49), and the highest prevalence of diarrheal disease was found in South Africa, 67.3% (95% CI: 63.21–71.38) (15), South Sudan,64.2%(95%CI:59.71-68.68) (16), Uganda,62.4%(95%CI:57.55-67.25) (14) and Ethiopia,41.75%(95%CI:40.53-42.97) (34)(Figure 2).

It would be beneficial to include a tree plot alongside Figure 2, presenting subgroup names and the number of studies in each group.

Please re-check, the overall score is not the same. Figure 2 23%, figure 3 16%. Is this because there’s double citation: Thiam 2017 in figure 2?

There should be table presentation like this after table 1, for example:

Area Pooled prevalence 95% CI

All Africa 7.5% 5.76 - 9.24

South Africa 67.3% 63.21 – 71.38

Etc.

Comment accepted and revision is done. The overall prevalence in Africa and regional sub group pooled prevalence is estimated using a forest plot( Figure 2).

Dear Reviewers, this data is directly taken from Table 1 as it is extracted from the source no additional analysis was done for this purpose.

Constructing this recommended table is best, but it already includes the pooled prevalence of DD in all of Africa, and each region is presented in forest plot in Figure 2. (If needed, I will do it in the next comment for additional revision).

The comment is accepted, and the tree plot alongside Figure 2 is done, including subgroup analysis by region as East Africa(EA), West Africa(WA), North Africa(NA), South Africa (SA), and Central Africa ( CA).

9 Table 2 The Df is empty Please add the Df number

Comment accepted and degree freedom is corrected(2)

11 Incongruence number of reported studies and in the factors table data

In the results, there were 8 studies mentioned. But in the Table, there are 10 studies data for poor hand washing factor.

Please recheck the number of studies in each subgroup, as well as the number of studies for other factors. Comment accepted and correction is done as it found in the table, there is nine studies (there was one study was double (Nantege et al). So the correct one is nine studies and the correction is done with nine studies( In table 4 ).

14 & 15 Factors order Table Example: in the table, Poor solid and poor liquid management were separated by six factors

It had better to rearrange the order so that poor solid and poor liquid waste management is closed enough for readability. Please check others e.g. breast feeding, etc. The comment is accepted, and the revision is done.

16 Discussion Paragraph 1 At least 3-5 sentences in one paragraph. Please discuss the Africa regional prevalence as well.

“The prevalence in… (Example South Africa) was….than in…. )

Discuss as well the contribution of the study findings for the research area, example:

This study updates the existing evidence on the prevalence of…… among …over the past decade, as well as identifying the various risk factors….

The comment is accepted, and the revision is done( paragraph 1,2 and 3.)

18 Discussions paragraph 1&3 Wordy If possible, please make the sentences more concise.

Comment accepted and revision is done.

Reviewer 2# Comments and suggestions

1 Paragraph section The paper is a systematic review that focuses on diarrheal disease in children under the age of five in Africa from January 2013 to December 2023 using PRISMA guidelines. However, a lot must be done to make this ready for publication Comment accepted and revision is done(cross check clean with truck changed manuscript).

2 Abstract result section The result section of the abstract needs to be rewritten for clarity. Improve sentence structure for readability. Comment accepted and revision is done

Using a systematic search of PubMed, Scopus, Web of Science, and Google Scholar for studies from 2013–2023, following PRISMA guidelines" Is ambiguous. Kindly rewrite this part for clarity. Comment is accepted and it is revised as: Search databases including PubMed, Scopus, Web of Science, and Google Scholar were used to find the research from January 2013 to December 2023. The PRISMA flow chart guideline was used to screen the research.

Result section There is no consistency in the font size usage and alignment of the paper. Some of the words are merged, such as Sudan,64.2%(95%CI:59.71-68.68) (16), Uganda,62.4%(95%CI:57.55-67.25) (14) and Ethiopia ,41.75%(95%CI:40.53-42.97)(34)(Figure 2) in Prevalence of Diarrheal disease among under-five children in Africa. Comment accepted and revision is done.

The degree of freedom in Cochran's Q statistics for heterogeneity is missing for Central Africa. Comment accepted and corrected df which was 2.

The dataset is dominated by studies from East Africa, with comparatively fewer studies from Central, Northern, and Southern Africa. This regional imbalance may skew the findings. Discussing how this affects generalizability would strengthen the conclusions. Comment accepted, considering this we author try to conduct sub group analysis by region and indicate pooled prevalence for each region.

The inclusion criterion limiting studies to those published in English may have excluded relevant research from French-speaking or Arabic-speaking African countries. This should be acknowledged. Comment accepted and acknowledged in limitation section.

The heterogeneity (I² = 99.3%) is relatively high, which raises concern about the comparability of studies included in the review. Consider explaining how the high heterogeneity might impact the pooled prevalence estimates and associated risk factors. Dear reviewer, considering this, sub group analysis were done and meta regression also done sample size, year of publication and the region with standard error take as covariate.

The plots are too compressed, especially the Sensitivity analysis and Funnel plots. Kindly revise all the plots to make them publication-worthy. The funnel plots suggested publication bias, which needs to be addressed. Dear reviewers due to included studies were large (66) sensitivity and Funnel plot is compressed. We authors try to modify the funnel plot and sensitivity analysis. Then the revised one is present better even the included study is large (66) for this purpose.

---

## [Decision Letter · Decision Letter 1]

Dear Dr. Worede,

Thank you for submitting your manuscript to PLOS ONE. After careful consideration, we feel that it has merit but does not fully meet PLOS ONE’s publication criteria as it currently stands. Therefore, we invite you to submit a revised version of the manuscript that addresses the points raised during the review process.

We look forward to receiving your revised manuscript.

Kind regards,

Hope Onohuean, PhD

Academic Editor

PLOS ONE

Journal Requirements:

Reviewers' comments:

Reviewer's Responses to Questions

**Comments to the Author**

Reviewer #1: All comments have been addressed

Reviewer #2: All comments have been addressed

2. Is the manuscript technically sound, and do the data support the conclusions?

Reviewer #1: Yes

Reviewer #2: Yes

3. Has the statistical analysis been performed appropriately and rigorously?

Reviewer #1: Yes

Reviewer #2: Yes

4. Have the authors made all data underlying the findings in their manuscript fully available?

Reviewer #1: Yes

Reviewer #2: Yes

5. Is the manuscript presented in an intelligible fashion and written in standard English?

Reviewer #1: Yes

Reviewer #2: Yes

Reviewer #1: 1. Please also indicate in the Methods section that you performed a meta-regression to quantify additional publication bias and conducted a sensitivity analysis to assess the robustness of your findings.

2. In the Results section for subgroup analysis, after presenting the prevalence of each subgroup, clearly state whether there is a significant or non-significant difference among them. Based on the p-value provided, it appears there is a significant difference. Offer a concise interpretation of these differences. For example:

“Subgroup analysis by region revealed the highest prevalence in [Region A], followed by [Region B], [Region C], and [Region D]. There was a significant difference among these regions (p-value = …), indicating that each region exhibits distinct [example: characteristics/outcomes].”

3. Please provide the DOI or URL for all references, along with the PubMed ID if available.

4. The year for Danquah’s reference [67] is missing. Kindly re-check all references to ensure accuracy of their publication years.

5. On pages 14, 16, 17, 19, and 20, replace “Kefalew et al.” with “Alemayehu.”

6. On page 9, Nwokoro is cited as reference [57]; on page 10, Nwokoro is cited as reference [63]; and on page 11, Nwokoro is cited as reference [82]. Please verify that the reference numbering is consistent throughout the manuscript.

7. For references 69/61, confirm whether the author’s name is “Tampah-Naah” or “Naah.” Please ensure consistency in author names on pages 23–34 and 30–37.

8. On page 11, use a consistent font to report the meta-regression results.

9. If feasible, consider including the PRISMA 2012 or PRISMA 2020 checklist as an additional file.

10. Good luck!

Reviewer #2: Our initial recommendations have mainly been resolved, but I found other minor issues that need to be addressed.

“Searches were conducted in databases including PubMed, Scopus, Web of Science, and Google Scholar to identified research ….” should be changed to “identify” in the method section.

“exclusive breastfeeding (AOR = 3.06, 95%CI: 2.12-4.43)” There is still no consistency in writing the figures as seen in ventilated improved pit latrines (AOR = 0.85, 95% CI: 0.77-0.95). Please keep to a consistent format of writing the “95% CI”.

“Regarding the study setting, 56 articles were done community-based…” should either be “56 articles were community-based” or “56 articles were done in community-based.

The reference at 53 and 54 are the same. Is there any reason why they are separated?

“The pooled prevalence of diarrheal disease among under-five children in Africa was found 23.590%” should be corrected.

In your discussion, you said, “Furthermore, a meta-analysis of five and six studies revealed that households with more than two under-five children and those with larger family sizes had a higher likelihood of developing DD compared to households with fewer children and smaller family sizes”. Why the choice of five and six studies? Can you add the reference to the studies used in that sentence?

**Do you want your identity to be public for this peer review?** For information about this choice, including consent withdrawal, please see our Privacy Policy

Reviewer #1: No

Reviewer #2: No

---

## [Author Response · Author response to Decision Letter 2]

5 May 2025

Dear Editor and Reviewers

Please accept our revised manuscript and note our point-by-point response to the reviewer below for the manuscript titled ‹‹ Prevalence of Diarrheal Diseases and Associated Factors among Under Five Children in Africa: A Meta-analysis››.

Our revised manuscript continues to meet the journal’s formal requirements, including the abstract and overall word count.

We look forward to your reply and decision.

Thank you so much for your comments and constructive feedback.

Version: 2

Date: 5th April/2025

Manuscript Number: PONE-D-24-52714R1.

Author response to Reviewer #1:

1. Please also indicate in the Methods section that you performed a meta-regression to quantify additional publication bias and conducted a sensitivity analysis to assess the robustness of your findings.

Author reflection: Comment accepted, and revision is done as A meta-regression analysis was also done to quantify additional publication bias. However, a sensitivity analysis was presented before this comment, which is «Sensitivity analyses were conducted to explore the robustness of the findings by excluding low-quality studies and studies with extreme prevalence estimates» (in the statistical analysis section, the last three lines).

2. In the Results section for subgroup analysis, after presenting the prevalence of each subgroup, clearly state whether there is a significant or non-significant difference among them. Based on the p-value provided, it appears there is a significant difference. Offer a concise interpretation of these differences. For example:

“Subgroup analysis by region revealed the highest prevalence in [Region A], followed by [Region B], [Region C], and [Region D]. There was a significant difference among these regions (p-value = …), indicating that each region exhibits distinct [example: characteristics/outcomes].”

Authors Reflection: Comment accepted and revision is done.

2. Please provide the DOI or URL for all references, along with the PubMed ID if available.

Authors Reflection: Dear reviewer thank you again for your constructive feedback. However, some papers not provid their DOI number and we try to present appropraite reference using Endote reference manager and hyperlink is done for all references.

3. The year for Danquah’s reference [67] is missing. Kindly re-check all references to ensure the accuracy of their publication years.

Author Refelection: Dear editors, thank you for your observation, Danquah’s publication year is 2014 and is checked.

4. On pages 14, 16, 17, 19, and 20, replace “Kefalew et al.” with “Alemayehu.”

Authors’ reflection: Dear Reviewers, thank you very much for your detailed observation, regarding your comments, there are two research papers in the author name start with Mulusew Alemayehu and Kefalew Alemayehu (Reference numbers 45 and 55 respectively). Therefore, we authors write this as Alemayehu M. et al and Kefalew A. et al) for better presentation and correction throughout the paper (Table 1 and 4).

5. On page 9, Nwokoro is cited as reference [57]; on page 10, Nwokoro is cited as reference [63]; and on page 11, Nwokoro is cited as reference [82]. Please verify that the reference numbering is consistent throughout the manuscript.

Authors Refelection: Yes, the comment is correct as it was in the revised truck changed manuscript, however, it was corrected in the clean manuscript in the previous reviewer's comments (because this research paper was written three times wrongly and was corrected in the previous comments) and currently only one Nwokoro author is present with the reference number of 57.

6. For references 69/61, confirm whether the author’s name is “Tampah-Naah” or “Naah.” Please ensure consistency in author names on pages 23–34 and 30–37.

Author Reflection: Thank you for your interesting comment. The correct author name is “Tampah-Naah,” and it has been used consistently throughout the revised manuscript. The reference is listed as [61] in the current clean version, and consistency has been ensured across all sections, including Tables 1 and 4.

7. On page 11, use a consistent font to report the meta-regression results.

Author reflection: Comment accepted and revision is done (font size is corrected 12 and New Times Roman,(Check the clean and truck change manuscript).

8. If feasible, consider including the PRISMA 2012 or PRISMA 2020 checklist as an additional file.

Author Refelection: Yes, the comment is acceptable, due to the attached additional file being more and we authors try to present this one in research paper screening process ( Figure1 PRISMA follow diagram).

9. Good luck!

Dear reviewers, we authors would like to thank you for your time and constructive feedback.

Author Response to Reviewer #2 Comments:

Our initial recommendations have mainly been resolved, but I found other minor issues that need to be addressed.

“Searches were conducted in databases including PubMed, Scopus, Web of Science, and Google Scholar to identified research ….” should be changed to “identify” in the method section.

Authors Reflection: Comment accepted and correction is done (check it in the abstract method section)

“exclusive breastfeeding (AOR = 3.06, 95%CI: 2.12-4.43)” There is still no consistency in writing the figures as seen in ventilated improved pit latrines (AOR = 0.85, 95% CI: 0.77-0.95). Please keep to a consistent format of writing the “95% CI”.

Author’s reflection: Comment accepted and revision is done (95%CI to 95% CI that is space is added consist for others)

“Regarding the study setting, 56 articles were done community-based…” should be either “56 articles were community-based” or “56 articles were done in community-based.

Author’s reflection: Comment accepted and revision is done as «56 articles were done in community-based».

The reference at 53 and 54 are the same. Is there any reason why they are separated?

Author reflection: Thank you for your observation. You are correct; the references at 53 and 54 were the same. This duplication was previously addressed and corrected in response to earlier reviewer comments, and the correct citation now appears as reference number 49: Akinyemi YC. Spatial pattern and determinants of diarrhea morbidity among under-five-aged children in Lagos State, Nigeria. Cities & Health. 2022;6(1):180–91. The duplicate has been removed accordingly.

“The pooled prevalence of diarrheal disease among under-five children in Africa was found 23.59%” should be corrected.

Author Reflection: dear Reviewers, thank you this pooled prevalence is correct, the result of the pooled prevalence of 66 research papers which s 25.59% and found in Figure 2. Because no research paper is added or removed in the current review process and no need for revision this pooled prevalence of DD.

In your discussion, you said, “Furthermore, a meta-analysis of five and six studies revealed that households with more than two under-five children and those with larger family sizes had a higher likelihood of developing DD compared to households with fewer children and smaller family sizes”. Why the choice of five and six studies? Can you add the reference to the studies used in that sentence?

Author Reflection: Thank you for your insightful comment. The selection of five and six studies was based on the number of included studies that specifically identified having more than two under-five children and larger family size as significant factors associated with DD. We authors have included those references in this section (discussion) ( you can see or check the truck change manuscripts)

---

## [Decision Letter · Decision Letter 2]

Prevalence of Diarrheal Diseases and Associated Factors Among Under Five Children in Africa: A Meta-analysis

PONE-D-24-52714R2

Dear Dr. Worede,

We’re pleased to inform you that your manuscript has been judged scientifically suitable for publication and will be formally accepted for publication once it meets all outstanding technical requirements.

Kind regards,

Hope Onohuean, PhD

Academic Editor

PLOS ONE

Additional Editor Comments (optional):

Reviewers' comments:

Reviewer's Responses to Questions

**Comments to the Author**

Reviewer #1: All comments have been addressed

2. Is the manuscript technically sound, and do the data support the conclusions?

Reviewer #1: Yes

3. Has the statistical analysis been performed appropriately and rigorously?

Reviewer #1: Yes

4. Have the authors made all data underlying the findings in their manuscript fully available?

Reviewer #1: Yes

5. Is the manuscript presented in an intelligible fashion and written in standard English?

Reviewer #1: Yes

Reviewer #1: Thank you for addressing all comments. Additional comment: after et al whether to use punctuation or not should be consistent. Some used (.), some do not use (.) punctuation. But I think the publisher will handle this.

**Do you want your identity to be public for this peer review?** For information about this choice, including consent withdrawal, please see our Privacy Policy

Reviewer #1: No

---

## [Editor Report · Acceptance letter]

PONE-D-24-52714R2

PLOS ONE

Dear Dr. Worede,

I'm pleased to inform you that your manuscript has been deemed suitable for publication in PLOS ONE. Congratulations! Your manuscript is now being handed over to our production team.

Kind regards,

on behalf of

Dr. Hope Onohuean

Academic Editor

PLOS ONE